# Innate immune activation by checkpoint inhibition in human patient-derived lung cancer tissues

Teresa WM Fan[1,2,3]*, Richard M Higashi[1,2,3], Huan Song[1,2,3], Saeed Daneshmandi[1,2,3], Angela L Mahan[3,4], Matthew S Purdom[3,5], Therese J Bocklage[3,5], Thomas A Pittman[6], Daheng He[3,7], Chi Wang[3,7], Andrew N Lane[1,2,3]*

[1]Center for Environmental and Systems Biochemistry (CESB), University of Kentucky, Lexington, United States; [2]Department of Toxicology and Cancer Biology, University of Kentucky, Lexington, United States; [3]Markey Cancer Center, University of Kentucky, Lexington, United States; [4]Departement of Surgery, University of Kentucky, Lexington, United States; [5]Departement of Pathology and Laboratory Medicine, University of Kentucky, Lexington, United States; [6]Department of Neurosurgery, University of Kentucky, Lexington, United States; [7]Department Internal Medicine, University of Kentucky, Lexington, United States

*For correspondence:
teresa.fan@uky.edu (TWMF);
andrew.lane@uky.edu (ANL)

Competing interest: The authors declare that no competing interests exist.

**Abstract** Although Pembrolizumab-based immunotherapy has significantly improved lung cancer patient survival, many patients show variable efficacy and resistance development. A better understanding of the drug's action is needed to improve patient outcomes. Functional heterogeneity of the tumor microenvironment (TME) is crucial to modulating drug resistance; understanding of individual patients' TME that impacts drug response is hampered by lack of appropriate models. Lung organotypic tissue slice cultures (OTC) with patients' native TME procured from primary and brain-metastasized (BM) non-small cell lung cancer (NSCLC) patients were treated with Pembrolizumab and/or beta-glucan (WGP, an innate immune activator). Metabolic tracing with $^{13}C_6$-Glc/$^{13}C_5$,$^{15}N_2$-Gln, multiplex immunofluorescence, and digital spatial profiling (DSP) were employed to interrogate metabolic and functional responses to Pembrolizumab and/or WGP. Primary and BM PD-1$^+$ lung cancer OTC responded to Pembrolizumab and Pembrolizumab + WGP treatments, respectively. Pembrolizumab activated innate immune metabolism and functions in primary OTC, which were accompanied by tissue damage. DSP analysis indicated an overall decrease in immunosuppressive macrophages and T cells but revealed microheterogeneity in immune responses and tissue damage. Two TMEs with altered cancer cell properties showed resistance. Pembrolizumab or WGP alone had negligible effects on BM-lung cancer OTC but Pembrolizumab + WGP blocked central metabolism with increased pro-inflammatory effector release and tissue damage. In-depth metabolic analysis and multiplex TME imaging of lung cancer OTC demonstrated overall innate immune activation by Pembrolizumab but heterogeneous responses in the native TME of a patient with primary NSCLC. Metabolic and functional analysis also revealed synergistic action of Pembrolizumab and WGP in OTC of metastatic NSCLC.

## Introduction

New checkpoint inhibitor-based immunotherapy, such as the use of programmed death one receptor (PD-1, CD279, or PDCD1) antibody Pembrolizumab (Pembro) has significantly improved the survival of lung and other cancer patients. However, even after stratification, a large fraction of the patient population shows variable efficacy, resistance development, and intolerable toxicity (**Arlauckas et al.,**

*2017*; *Ayers et al., 2017*; *Garon et al., 2015*; *Meyers et al., 2018*; *Morgensztern and Herbst, 2016*; *Rizvi et al., 2015*). Clearly better therapeutic prediction and understanding of the drug action are needed to improve patient care and outcomes. Functional heterogeneity of the tumor microenvironment (TME) is crucial to modulating drug responses including resistance development (*Dalton, 1999*; *Ostman, 2012*; *Qu et al., 2019*; *Trédan et al., 2007*; *Wang et al., 2009*). However, our understanding of individual patients' TME that impacts drug response is hampered by lack of appropriate models.

Multiple preclinical models have been established for basic and translational cancer research, ranging from 2D/3D cell cultures, 3D patient-derived organoids (PDO), ex vivo organotypic tissue slice cultures (OTC), to in vivo xenograft and patient-derived xenograft (PDX) animal models (*Fan et al., 2020*). Each model has its advantages and limitations. The use of tissue slices (with no culturing involved) was pioneered by O. Warburg who demonstrated enhanced lactic fermentation under aerobic conditions (*Warburg, 1924*), which is now recognized as a hallmark of cancers (*Hanahan and Weinberg, 2011*). We have adopted and improved this approach by culturing ex vivo lung tissue slices derived from NSCLC patients. We have shown that such ex vivo cultures recapitulate the metabolic reprogramming observed in vivo (*Sellers et al., 2015*). This model is versatile in experimental manipulations and retains patients' native 3D architecture/tissue microenvironment including infiltrated immune cells, which is not represented by multiply passaged mouse PDX or 3D cell models including PDO. Furthermore, the matched cancer (CA) and non-cancer (NC) tissue design provides the capacity for distinguishing on- and off-target drug actions in individual patients (*Lane et al., 2016*). We previously showed that two different PD lung OTC treated with the natural product immune activator (β-glucan) displayed distinct responses of macrophage (MΦ) metabolism that was related to variable tissue damage (*Fan et al., 2016*).

It is now widely recognized that metabolic reprogramming is crucial to immune cell activation to meet the demand for energy/anabolic metabolism (*Gnanaprakasam et al., 2017*; *O'Neill and Pearce, 2016*; *Palmieri et al., 2020*) and specific immunomodulatory events. Notably, succinate production in MΦ stabilized HIF1α, resulting in the release of proinflammatory IL-1β (*Tannahill et al., 2013*). Tryptophan (Trp) catabolism to quinolinate fuels de novo $NAD^+$ synthesis and blockade of this pathway led to a pro-inflammatory state in aged MΦ (*Minhas et al., 2019*). Reprogrammed metabolic events in the TME can also favor tumor immune escape (*Chang et al., 2015*; *Fischer et al., 2007*; *Ganeshan and Chawla, 2014*; *Ghesquière et al., 2014*; *Lim et al., 2017*; *Sica and Bronte, 2007*). For example, enhanced glycolysis (e.g. induced by the expression of checkpoint ligand PD-L1) in cancer cells can deplete glucose leading to T-cell 'anergy' in the TME (*Chang et al., 2015*; *Fischer et al., 2007*; *Lim et al., 2017*). The resulting excess release of lactate and protons into the TME can also deactivate infiltrating MΦ and T cells (*Fischer et al., 2007*; *Gurusamy et al., 2017*; *Munn and Bronte, 2016*; *Parker et al., 2015*). The expression of PD-L1 receptor, PD-1 was also shown to negatively regulate T-cell function and survival by modulating T-cell metabolism (*Bettonville et al., 2018*; *Patsoukis et al., 2015*; *Tkachev et al., 2015*). Moreover, tumor indoleamine 2,3-dioxygenase (IDO) depleted Trp in the TME by oxidation to kynurenine, which led to T-cell inhibition (*Joyce and Fearon, 2015*; *Munn and Mellor, 2013*). All of these immunomodulatory events were primarily learned from animal models or in vitro human cell cultures. The question is whether they are recapitulated in the patient tumor tissues.

To address this question, we prepared ex vivo lung OTC procured from one each patient with primary and brain-metastasized (BM) NSCLC. They were subject to Pembro ± β-glucan treatments with $^{13}C_6$-glucose or $^{13}C_5,^{15}N_2$-Gln as tracers for tracking perturbations in metabolic networks by stable isotope-resolved metabolomic (SIRM) analysis (*Fan et al., 2016*; *Fan et al., 2019*; *Sellers et al., 2015*). We found that Pembro activated the pentose phosphate pathway (PPP), glycogen turnover, and Gln uptake but blocked Gln use in the Krebs cycle, itaconate synthesis, and quinolinate buildup in the primary NSCLC OTC. For the metastatic NSCLC OTC, Pembro or β-glucan alone had little metabolic effect but the combination treatment suppressed the PPP, glycogen synthesis, and nucleotide synthesis. Reduced mitotic index and significant tissue damage were evident in Pembro-treated primary and Pembro + β-glucan-treated BM-NSCLC OTC. Metabolic changes in the primary OTCs were correlated with the release of pro-inflammatory effectors and reduced expression of cancer cell/anti-inflammatory MΦ markers. These changes point to innate immune activation by Pembro, but different TMEs responded variably as shown by digital spatial profiling (DSP) and/or immunofluorescence analysis.

## Results and discussion

### PD-1$^+$, CD8$^+$ T cells are present in both primary and metastatic NSCLC tumors

We performed multiplex immunofluorescent (mIF) staining on the primary (from patient UK131) and metastatic (from patient UK2035) NSCLC tissues to characterize the tumor cell type, the status of the checkpoint proteins, and immune cell properties (see *Supplementary file 1*, for staging and demographics). As shown in *Supplementary file 2*, the UK131 tumor stained positive for KRT5 (**A**), which is consistent with its clinical classification as lung squamous cell carcinoma (LSCC) (*Xiao et al., 2017*). Although the KRT5$^+$ cancer cells were positive for HIF1α, this staining occurred much more in the cytoplasm than in the nucleus, which suggests that the major fraction of HIF1α was not transcriptionally active. Also present were the smaller non-cancer HIF1α$^+$ cells, which are likely infiltrated immune cells (**A**). More notable was the presence of PD-1 expressing CD8$^+$ T cells (**B**), which indicates a high likelihood of this tissue to respond to Pembro treatment. Also evident were PD-L1 expressing CD206$^+$ cells (**C**), which are characteristic of tumor-associated MΦ (TAM) (*Van Overmeire et al., 2014*) and reflective of an immunosuppressive TME (*Lu et al., 2019*).

Positive mIF staining of chromogranin A (CgA) and NCAM1 (*Supplementary file 3*) confirmed the clinical classification of the metastatic UK2035 tumor as a lung neuroendocrine tumor (*Kashiwagi et al., 2012*). PD-1 expressing CD8$^+$ T cells and PD-L1$^+$ cells were also present in this tumor, with the former cells far fewer in numbers than the latter cells (*Supplementary file 3*).

### Primary tumor tissue slice cultures of UK131 patient respond to Pembro treatment

#### Pembro activates the pentose phosphate pathway and glycogen synthesis in UK131 CA OTC

We treated the matched pair of CA and NC lung OTC of patient UK131 with Pembro for 24 hr in the presence of $^{13}C_6$-glucose ($^{13}C_6$-Glc) or $^{13}C_5$,$^{15}N_2$-Gln. We found that the CA OTC responded to Pembro with a metabolic shift. Namely, Pembro enhanced $^{13}C_6$-Glc-derived $^{13}C$ labeling in most of the PPP metabolites compared to the control (Ctl) (*Figure 1*). These included $^{13}C_6$-glucose-6-phosphate (G6P) (**6, a**), $^{13}C_5$-ribulose/ribose-5-phosphate (R5P) (**5, c**), $^{13}C_7$-sedoheptulose-7-phosphate (S7P) (**7, d**), $^{13}C_4$-erythrose-4-phosphate (E4P) (**4, e**), and $^{13}C_6$-fructose-6-phosphate (F6P) (**6, f**), but not $^{13}C_6$-6-phosphogluconate (6 PG) (**6, b**). These data point to activation of the PPP with a partial block at the glucose-6-phosphate dehydrogenase (G6PD) step in the oxidative branch. As such, the non-oxidative (Nonox) branch reactions catalyzed by the transketolase (TK) and transaldolase (TA) contributed more to the $^{13}C$ incorporation into R5P, S7P, E4P, and F6P in Pembro versus Ctl-treated CA OTC. This agrees with the Pembro-induced higher buildup of scrambled $^{13}C$ isotopologues (**Scr**) of these metabolites (*Figure 1*). In addition to PPP activation, Pembro enhanced $^{13}C$ labeling of glycogen (**i**) and its direct precursor UDP-glucose (UDPG, **h**) (*Figure 1*), which suggests stimulation of glycogen synthesis by Pembro. None of these changes were evident in the matched NC OTC, which indicates that these effects are tumor specific.

#### Pembro blocks Gln oxidation through the Krebs cycle in UK131 CA lung tissues

With $^{13}C_5$,$^{15}N_2$-Gln as tracer, Pembro moderately increased $^{13}C$-Gln uptake in CA OTC, compared with the Ctl. This was accompanied by enhanced release of $^{13}C$-Glu and $^{13}C$-succinate into the medium (*Figure 2A–a*), which could at least in part account for Pembro-induced increase in Gln uptake. In comparison, there were only minor changes in the glucose uptake and lactate released into the medium (*Figure 2A–b*). Thus, Pembro enhanced glutaminolysis (Gln conversion to Glu and succinate) but had a negligible effect on glycolysis. We also tracked intracellular oxidation of glutaminolytic products via the Krebs cycle (*Figure 2B*). In CA OTC, Pembro modestly depleted the first two products $^{13}C_5$-Glu (**C5Nx, a**)/ $^{13}C_5$-αKG (**C5, b**) and subsequent Krebs cycle products ($^{13}C_4$- and other $^{13}C$-isotopologues of succinate, fumarate, malate, citrate, and Asp) (**c–g**). $^{13}C$ labeling of glutathione (GSH, **h**) was less affected by the Pembro treatment. Blockade of the glutaminase (GLS) activity and/or diversion of Glu to GSH metabolism could account for these Pembro-induced changes in the $^{13}C$ labeling patterns of tissue metabolites. In addition, GSH synthesis requires cystine uptake from the medium in exchange

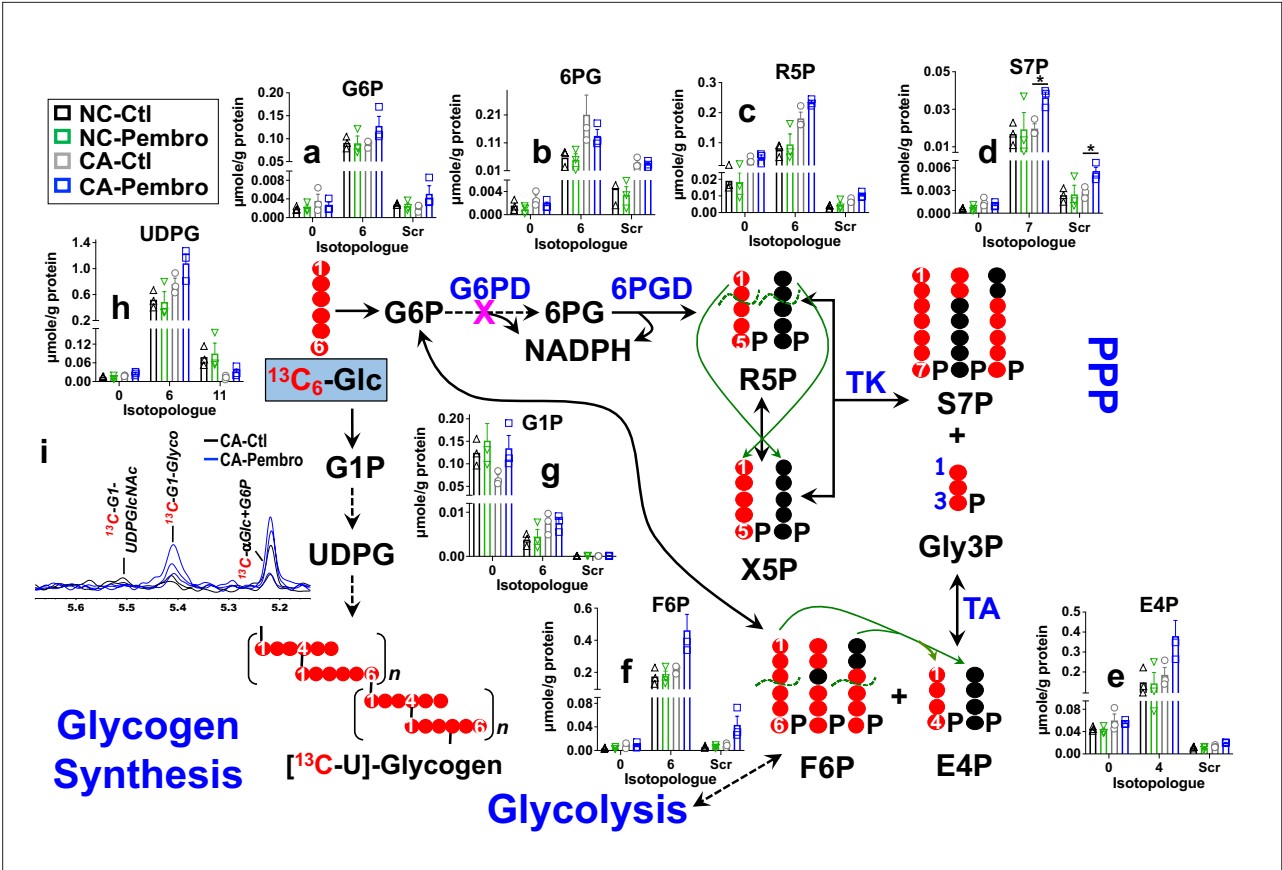

**Figure 1.** Pembro activates the PPP and glycogen synthesis in primary CA, but not in NC lung tissues. Matched pairs of fresh cancerous (CA) and non-cancerous (NC) lung tissue slice cultures of UK131 patient were treated with 40 μg/mL Pembro + $^{13}C_6$-glucose ($^{13}C_6$-Glc) for 24 hr before extraction for polar metabolites, as described in the Experimental. The diagram traces the atom-resolved $^{13}C_6$-Glc transformation through the PPP and glycogen synthesis, the green arrows and dashed curves delineate the reactions of transketolase (TK) and transaldolase (TA) that scramble $^{13}C$ labeling in PPP metabolites, and **X** denotes the block at glucose-6-phosphate dehydrogenase (G6PD) step. Panels (**a–h**) were obtained from IC-UHR-FTMS analysis and panel (**i**) from 1D HSQC NMR analysis (n = 3). X-axis denotes the number of $^{13}C$ or scrambled $^{13}C$ (Scr) in each metabolite. ●: $^{13}C$; ●: $^{12}C$; ■: NC control (Ctl); ■: NC Pembro; ■: CA Ctl; ■: CA Pembro; G6P: glucose-6-phosphate; 6 PG: 6-phosphogluconate; R5P: ribulose/ribose-5-phosphates; S7P: sedoheptulose-7-phosphate; Gly3P: glyceraldehye-3-phosphate; E4P: erythrose-4-phosphate; F6P: fructose-6-phosphate; G1P: glucose-1-phosphate; UDPG: UDP-glucose; Glyco: glycogen; UDPGlcNAc: UDP-N-acetylglucosamine. Panels (**a–h**) show mean ± sem (n = 3). *p<0.05.

The online version of this article includes the following figure supplement(s) for figure 1:

**Figure supplement 1.** M1 polarization of human MΦ activates the PPP and glycogen synthesis.

for Glu release via the xCT antiporter (**Shih et al., 2006**), which would contribute to the buildup of $^{13}C$-Glu in the medium and add to the depletion of tissue $^{13}C$-Glu. Likewise, increased release of $^{13}C$-succinate to the medium would further deplete tissue succinate. Again, none of these changes were evident in the matched NC counterparts.

## Pembro-induced metabolic shift in CA OTC is akin to that induced by M1-type polarization of human MΦ

Activation of the PPP and glycogen metabolism is known to occur during proinflammatory or M1-type polarization of murine MΦ and model human MΦ THP-1 cells by lipopolysaccharides (LPS) (**Ma et al., 2020**; **Nagy and Haschemi, 2015**). PPP and glycogen metabolism via PPP (**Ma et al., 2020**) help sustain NADPH production in activated MΦ to maintain redox homeostasis and fuel the demand for lipid and deoxynucleotide synthesis. To verify such metabolic shift in activated human MΦ, we isolated peripheral blood monocytes (PBMC) from a male volunteer age-matched to the cancer patient, followed by differentiation into the M0 state and treatment with LPS+ interferon-gamma (IFNγ) for classical M1 polarization or with IL-4+ IL-13 for alternative M2 polarization in the presence of

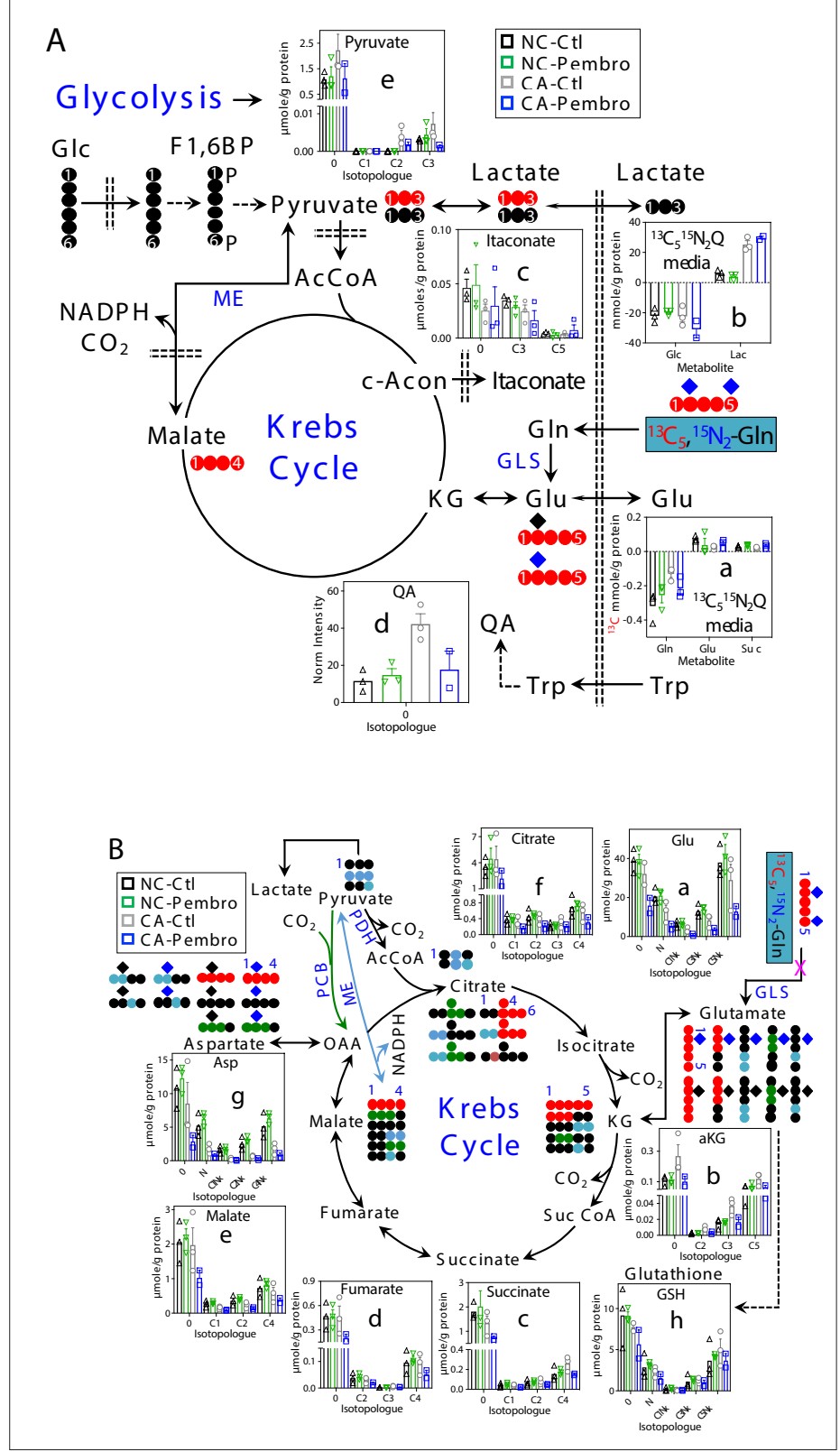

**Figure 2.** Pembro enhances Gln uptake/release of glutaminolytic products while blocking Gln-fueled Krebs cycle activity in primary CA but not in NC lung tissues. A separate set of CA and NC lung tissue slices from UK131 was treated with 40 μg/mL Pembro plus $^{13}C_5,^{15}N_2$-Gln ($^{13}C_5^{15}N_2Q$) for 24 hr before extraction and analysis for polar metabolites as in *Figure 1*. Diagram in (**A**) depicts the glutaminolytic production of $^{13}C$-Glu and release into the

*Figure 2 continued on next page*

*Figure 2 continued*

medium, $^{13}C_5$-Glu transformation through the Krebs cycle, and $^{13}C_3$-pyruvate/lactate production from the malic enzyme (ME) pathway, $^{13}C$-itaconate production from c-aconitate (c-Acon), Trp uptake and oxidation to quinolinate (QA) as well as unlabeled lactate production from glucose (Glc) via glycolysis. The diagram in (**B**) traces atom-resolved transformation of $^{13}C_5,^{15}N_2$-Gln through the Krebs cycle (both PDH- ● and PCB ● -initiated) and ME (●) pathway. Panels (**a, b**) in (**A**) were obtained from 1D HSQC analysis while the rest of all panels from IC-UHR-FTMS analysis. Numbers in X-axis denote those of $^{13}C$ while x indicates 1–2 $^{15}N$ and N is $^{15}N_1$. ●: $^{12}C$; ◆: $^{14}N$; ◆: $^{15}N$; F1,6BP: fructose-1,6-bisphosphate; Lac: lactate; AcCoA: acetyl coenzyme A; αKG: α-ketoglutarate; Suc: succinate; Suc CoA: succinyl CoA; GSH: glutathione; GLS: glutaminase; PDH: pyruvate dehydrogenase; PCB: pyruvate carboxylase. Panels a-h display mean ± sem, with n = 3.

The online version of this article includes the following figure supplement(s) for figure 2:

**Figure supplement 1.** M1 polarization of human MΦ attenuates glutaminolysis and subsequent oxidation through the Krebs cycle but activates itaconate and quinolinate production.

$^{13}C_6$-Glc. Comparing M1- versus M2-type MΦ, we saw very similar shift in the $^{13}C$ labeling patterns of the PPP and glycogen products (*Figure 1—figure supplement 1*) as those observed for the Pembro-versus Ctl-treated CA OTC (*Figure 1A*). Particularly noted were the break in G6P (**a**) to 6 PG (**b**) conversion and the enhanced buildup of $^{13}C$ scrambled products of PPP (**c–f**) in M1- versus M2-type MΦ (*Figure 1—figure supplement 1*); the latter is consistent with Nonox PPP being the major source of pentose phosphates in LPS-activated murine MΦ (*Haschemi et al., 2012*).

We also observed similarities in the shift of $^{13}C_5,^{15}N_2$-Gln metabolism induced by M1 polarization of human MΦ (*Figure 2—figure supplement 1*) compared with those elicited by Pembro in CA OTC (*Figure 2*). These included enhanced release of $^{13}C$-succinate (**A–a**), lack of effect on the excretion of glycolytic lactate, reduced Gln oxidation through glutaminolysis and the Krebs cycle (**B–a** to **g**), and lack of attenuation in GSH synthesis (**B–h**) (*Figure 2—figure supplement 1*). However, some differential responses were also evident, notably reduced $^{13}C$-Glu release (**A–a**) as well as increased buildup of

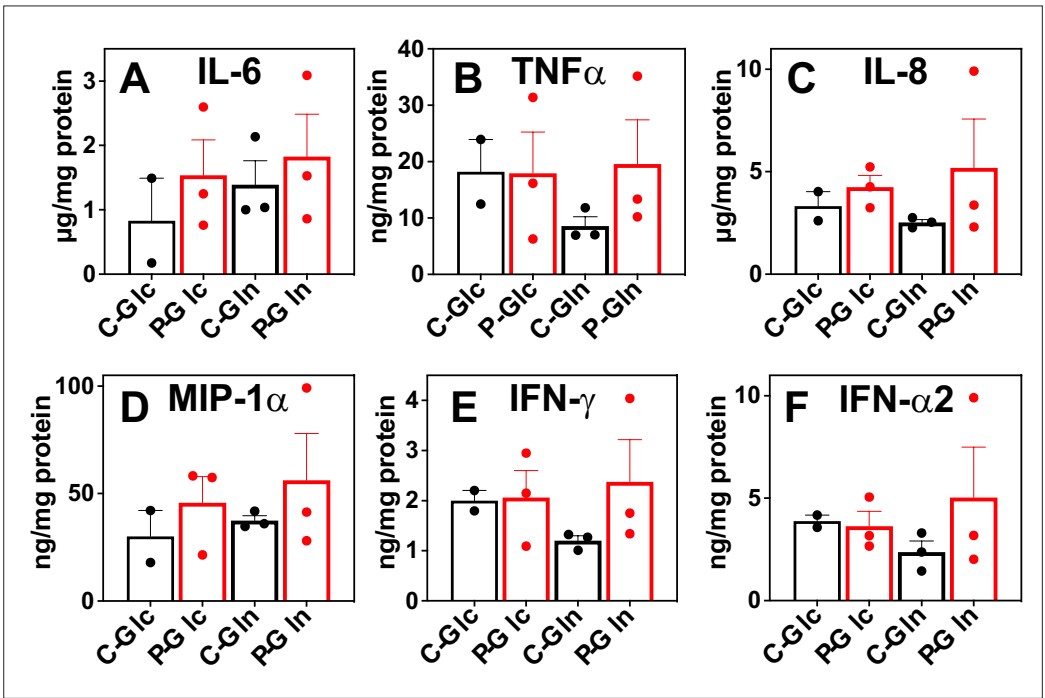

**Figure 3.** Pembro enhances the release of proinflammatory factors in slice cultures of NSCLC OTC from UK131 patient. Media (**A–F**) from the CA samples in *Figure 2* were analyzed for cytokines and chemokines as described in the Experimental. C-Glc or C-Gln (■): Ctl + $^{13}C_6$-Glc or $^{13}C_5,^{15}N_2$-Gln; P-Glc or P-Gln (■): 40 μg/mL Pembro + $^{13}C_6$-Glc or $^{13}C_5,^{15}N_2$-Gln. Mean ± se (n = 3) are displayed.

The online version of this article includes the following figure supplement(s) for figure 3:

**Figure supplement 1.** M1 polarization of human MΦ enhances the release of proinflammatory effectors.

<sup>13</sup>C-itaconate (**A–c**) and unlabeled quinolinate (QA, **A–d**) in M1 versus M2-type Mϕ (*Figure 2—figure supplement 1*) as opposed to the lack of or opposite effect observed for Pembro- versus Ctl-treated CA OTC (*Figure 2A*). These differences could reflect the influence of TME on MΦ metabolism and/or metabolic contribution of other cell types in the TME.

Besides metabolic shifts, we examined the culture media for secreted immune effectors. The release of proinflammatory cytokines/chemokines into the media was consistently enhanced by the Pembro treatment of CA OTC, as shown in *Figure 3A–F*. Some of these Pembro-induced changes were evident in NSCLC tumor PDOs (*Mediavilla-Varela et al., 2017*). Although the none of the Pembro effects on CA OTC reached statistical significance of p-value ≤ 0.05, consistent trend was seen that points to M1 type polarization. This is corroborated by the increased release of the same set of effectors (IL-6, TNFα [*Mercalli et al., 2013*], IL-8 [*Meniailo et al., 2018*], MIP-1α or CCL3 [*Maurer and von Stebut, 2004*], IFNγ, and IFNα2 [*Kumagai et al., 2007*]) in M1- versus M2-polarized human

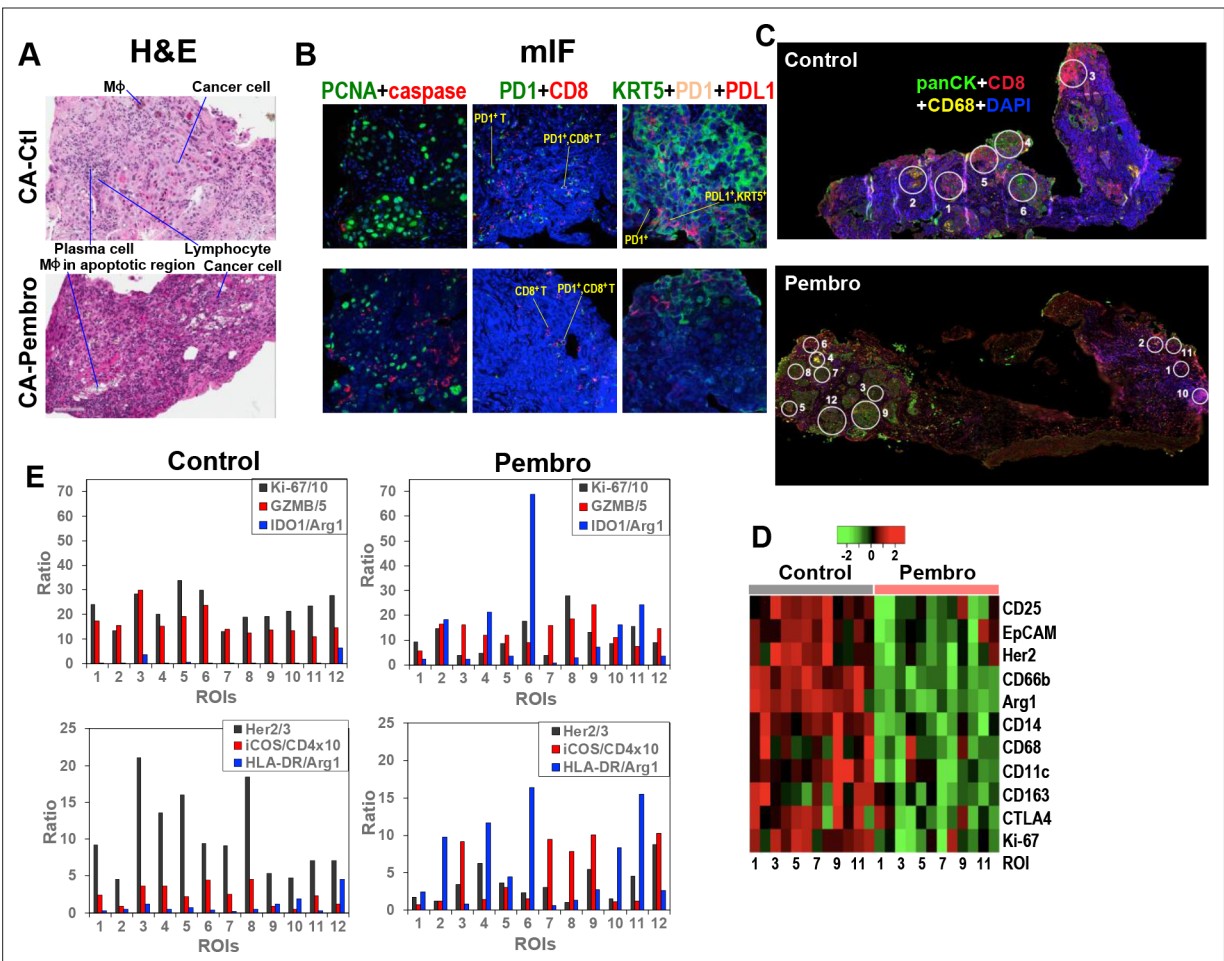

**Figure 4.** Heterogeneous cellular and functional responses of UK131 tumor tissue slice culture to Pembro. The treated tissue slices from *Figure 2* were subsampled, paraffin-embedded, and sectioned into 4 µm slices for H&E staining (**A**), mIF staining (**B**), and digital spatial profiling, as described in the Experimental. White circles in (**C**) are regions of interest (ROIs) in 100 and 200 µm in diameter, which were selected based on the abundance of cancer cells (panCytokeratin or panCK in green; e.g. 4, 6 in Control; 3, 7–8, 9, 12 in Pembro), CD8<sup>+</sup> T cells (CD8 in red; e.g. 3 in Control; 1, 10 in Pembro), MΦ (CD68 in yellow; e.g. 2 in Control; four in Pembro), and mixed cell populations (e.g. 1, 5 in Control; 2, 5, 11 in Pembro). ROIs 7–12 for control tissue was not shown. These ROIs were probed for the expression level of 58 different markers of cancer and immune cell functional states using Oligo-barcoded antibodies, as described in the Experimental. Eleven markers that were differentially expressed region-wide between Control and Pembro treatments are shown as heat map in (**D**). The scale in D is log₂. ROI-specific quantification of notable markers for Ctl- and Pembro-treated tissues is shown in (**E**), where the ratios displayed were marker-specific Oligo counts were normalized to the geometric mean of the house keeping genes GAPDH, S6, and Histone H3. Some ratios were multiplied or divided for display purposes.

MΦ (*Figure 3—figure supplement 1*). It is interesting to note that MIP-1 are chemokines that regulate both pro-inflammatory responses and homeostasis (wound healing; *Maurer and von Stebut, 2004*).

## Pembro-induced metabolic shifts are accompanied by CA tissue damage and TME-dependent immune responses

Histological analysis of Pembro-treated CA OTC sections by hematoxylin and eosin (H&E) staining revealed extensive tissue damages including the presence of MΦ in apoptotic regions, as shown in *Figure 4A*. mIF staining of separate sections of the same OTC confirmed increased apoptosis (red caspase fluorescence), which was accompanied by decreased mitotic index (green PCNA fluorescence) (*Figure 4B*). Also evident were the decrease in KRT5+/PD-L1+, KRT5+ cancer cell as well as PD-1+ and PD-1+/CD8+ T-cell populations. The CA OTC response is expected based on Pembro-mediated activation of exhausted PD-1+ T cells that possess anti-tumor activity (*Graves et al., 2019*).

To determine how the response of CA tissue varied with the TME, we analyzed another section of the same OTCs by DSP. DSP is a new technology that enables high-plex analysis of many protein markers in different regions of interest (ROIs) of FFPE tissue sections (*Zugazagoitia et al., 2020*). We chose 12 ROIs at 100 or 200 μm diameter for each Ctl and Pembro-treated tissue sections (*Figure 4C* and data not shown) as described previously (*Fan et al., 2020*). The choices were based on TME regions enriched in T cells (CD8+, red), MΦ (CD68+, yellow), cancer cells (panCK+, green), and combinations of the three cell types. ROI-dependent analysis of 58 protein markers revealed 11 markers whose expression was attenuated across most ROIs of Pembro-treated versus Ctl tissue sections (*Figure 4D*). These included Ki67 as well as cancer cell markers EpCAM and Her2. The decreased expression is consistent with tissue damage and reduced cancer cell population as observed by H&E and mIF imaging (*Figure 4A,B*). Also evident was the reduced expression of M2-type MΦ markers, CD163 and Arg1 as well as regulatory T (Treg) cell marker CD25. CD25+ Treg cells are known to induce M2-type polarization in co-cultured monocytes, as evidenced by the upregulation of CD163 and downregulation of HLA-DR expression (*Tiemessen et al., 2007*). Thus, CD25 suppression may be related to reduced CD163 expression in the Pembro-treated TME. Although there was no consistent increase in HLA-DR expression across ROIs, we saw an overall increase in the ratio of this M1–MΦ marker to the M2–MΦ marker Arg1 in Pembro- versus Ctl-treated CA tissues (*Figure 4E*). This was also the case for another human M1–MΦ marker IDO1. These data point to repolarization of M2- to M1-type MΦ in the Pembro-treated CA tissue. In contrast to the response of the Treg marker CD25, the ratio of a T-cell activation marker iCOS (*Dong et al., 2001*) to T-cell marker CD4 largely increased (*Figure 4E*) along with decreased expression of the checkpoint protein CTLA4 (*Figure 4D*) across all ROIs in response to Pembro. Reduced Treg cell occurrence was commonly noted in CA tissues from Pembro-treated cancer patients (*Kumar et al., 2020*). However, enhanced iCOS expression in T cells was not known to be induced by anti-PD-1 treatment (*Wei et al., 2017*). It should be noted that since no immune cell infiltration can occur in our OTC model, the Pembro-induced T-cell or MΦ marker response was derived from the resident populations only.

Moreover, we observed variable ROI-dependent distributions of immune functional and cancer cell markers in relation to the mitotic index marker Ki-67 in Pembro- and Ctl-treated CA tissues (*Figure 4E*). In particular for Pembro-treated tissues, panCK+ cancer cells-rich ROIs 3, 7, 9, and 12 had much higher granzyme B (GZMB) relative to Ki-67 levels than ROIs 6 and 8. ROIs 3, 7, 9, and 12 also displayed higher iCOS/CD4 ratios than ROIs 6 and 8. These data point to more activated T cells and reduced proliferative capacity of cancer cells in the former than the latter regions. We also noted attenuated expression of a lung cancer driver protein Her2 (*Garrido-Castro and Felip, 2013*) in ROIs 6/8 versus ROIs 3/7/9/12, which suggests a difference in cancer cell property between the two sets of ROIs. Moreover, ROI six displayed very high IDO1/Arg1 and HLA-DR/Arg1 ratios than other ROIs including ROI 8. This could reflect a particularly high polarization of TAM toward M1-type and/or IDO1 overexpression in cancer cells in ROI 6. In either case, cancer cells in this TME appeared to remain proliferative. As for ROI 8, the proliferative capacity was the highest with the lowest IDO1 or HLA-DR to Arg1 ratios but a comparable level of cytotoxic GZMB and ICOS/CD4 ratio to those of ROIs 3/7/9/12. These data implicate the presence of

Pembro-resistant cancer cells in ROIs 6 and 8, which can overcome M1 polarization in ROI six and T-cell activation in ROI 8. If ROI six cancer cells were high in IDO1 activity, this can deplete Trp in the TME while producing immunosuppressive metabolites such as kynurenine, both of which could contribute to immune evasion (*Li et al., 2019*; *Moffett and Namboodiri, 2003*) and thus cancer cell resistance. IDO1 overexpression occurred in NSCLC and higher serum kynurenine to Trp ratios/QA levels were associated with NSCLC disease progression (*Bianco et al., 2019*; *Botticelli et al., 2018*).

Our metabolomic data showed that QA built up in response to M1-MΦ activation (*Figure 2—figure supplement 1*), which presumably resulted from a block in the downstream enzyme quinolinate phosphoribosyl-transferase, which is involved in de novo $NAD^+$ synthesis (*Minhas et al., 2019*). Trp-fueled $NAD^+$ repletion is required for mitochondrial signaling and metabolism during immune resolution. QA depletion by Pembro in CA OTC (*Figure 2A–d*) is unlikely to result from M2-type MΦ polarization as Arg1 expression was down (*Figure 4D*). Instead, it is consistent with excess Trp consumption by IDO1-overexpressing cancer cells, thereby interfering with immune activation.

## Metastatic tumor OTCs of UK2035 patient respond to Pembro+WGP co-treatment

### Pembro+WGP blocks the Krebs cycle, pentose phosphate pathway, and glycogen/nucleotide synthesis in UK2035 CA lung OTCs

We treated ex vivo OTCs of BM NSCLC tissues of patient UK2035 with none (Ctl), Pembro, WGP, or Pembro + WGP (*P + W*) in the presence of $^{13}C_6$-Glc for 24 hr. We found no consistent changes in the $^{13}C$ labeling of the glycolytic and Krebs cycle intermediates or end products in response to Pembro (■) or WGP (■) treatment (**a**, **c–h**), except for the WGP-elicited reduction of F1,6BP (**b**) (*Figure 5A*). Nor were there consistent changes for GSH and itaconate derived from the Krebs cycle (**k–l**). However, *P + W* treatment (■) blocked $^{13}C$ incorporation into tissue F1,6BP, pyruvate (**c**), citrate (**e**), c-aconitate (**f**), Asp (**i**), and Glu (**j**) but not the uptake of $^{13}C$-Glc nor the release of $^{13}C$-Lac into media. These data suggest that *P + W* disrupted the first half of the Krebs cycle activity (PDH to IDH), but not glycolysis as a whole.

We then tracked $^{13}C$ incorporation into the products of the PPP and glycogen metabolism in treated UK2035 CA OTCs as shown in *Figure 5B*. The combined *P + W* treatment substantially depleted all major $^{13}C$ labeled products of PPP (**a–f**). Neither Pembro nor WGP treatment alone affected the $^{13}C$ labeling of the PPP products except for the depletion of $^{13}C$ scrambled 6 PG (**Scr**, **b**) by WGP. However, no depletion was evident for 6 PG's precursor, G6P (**a**) and downstream products (**c–f**) under WGP treatment. These data point to a block at G6PDH, which agreed with WGP's capacity for M1 repolarization as such block was also evident in human M1–MΦ (*Figure 1—figure supplement 1*). Moreover, *P + W* but not Pembro or WGP treatment alone attenuated the synthesis of $^{13}C$-glycogen (**i**) and its precursors $^{13}C_6$-G1P (**g**) and -UDPG (**h**) relative to the Ctl treatment (*Figure 5B*).

We further tracked the $^{13}C$ fate through glycolysis, the Krebs cycle, and PPP into nucleotide synthesis, as shown in *Figure 5C* for pyrimidine and *Figure 5D* for purine nucleotides and their derivatives. The *P + W*, but not Pembro or WGP treatment alone, attenuated the $^{13}C$ incorporation into the ribose (**5**, **d**) and pyrimidine base (**Base**, **d**) of UTP, which reflected reduced synthesis of the respective $^{13}C$-precursors phosphoribosyl pyrophosphate PRPP (**5**, **c**) and Asp (**a**) (*Figure 5C*). Thus, disruption of the Krebs cycle and the PPP at least in part accounted for decreased UTP synthesis.

Similarly, the incorporation of $^{13}C_6$-Glc-derived $^{13}C$-Gly and -ribose into ATP (**b**) and GTP (**c**) via the purine nucleotide synthesis pathway was attenuated by *P + W* but not by Pembro or WGP alone (*Figure 5D*). This was shown by the reduced levels of both $^{13}C_{base}$ and $^{13}C_5$ isotopologues of ATP and GTP in *P + W*-treated CA OTC. These were opposite to the buildup of the corresponding isotopologues of the precursor IMP (**a**), which points to a block downstream of IMP, possibly due to the reduced availability of the amino donor Asp (*Figure 5C–a*).

Altogether, *P + W* treatment inhibited energy and anabolic metabolism in metastatic NSCLC OTC, while Pembro or WGP treatment alone did not. These results point to a synergistic interaction of Pembro and WGP in disrupting central metabolism in UK2035 tumor tissues.

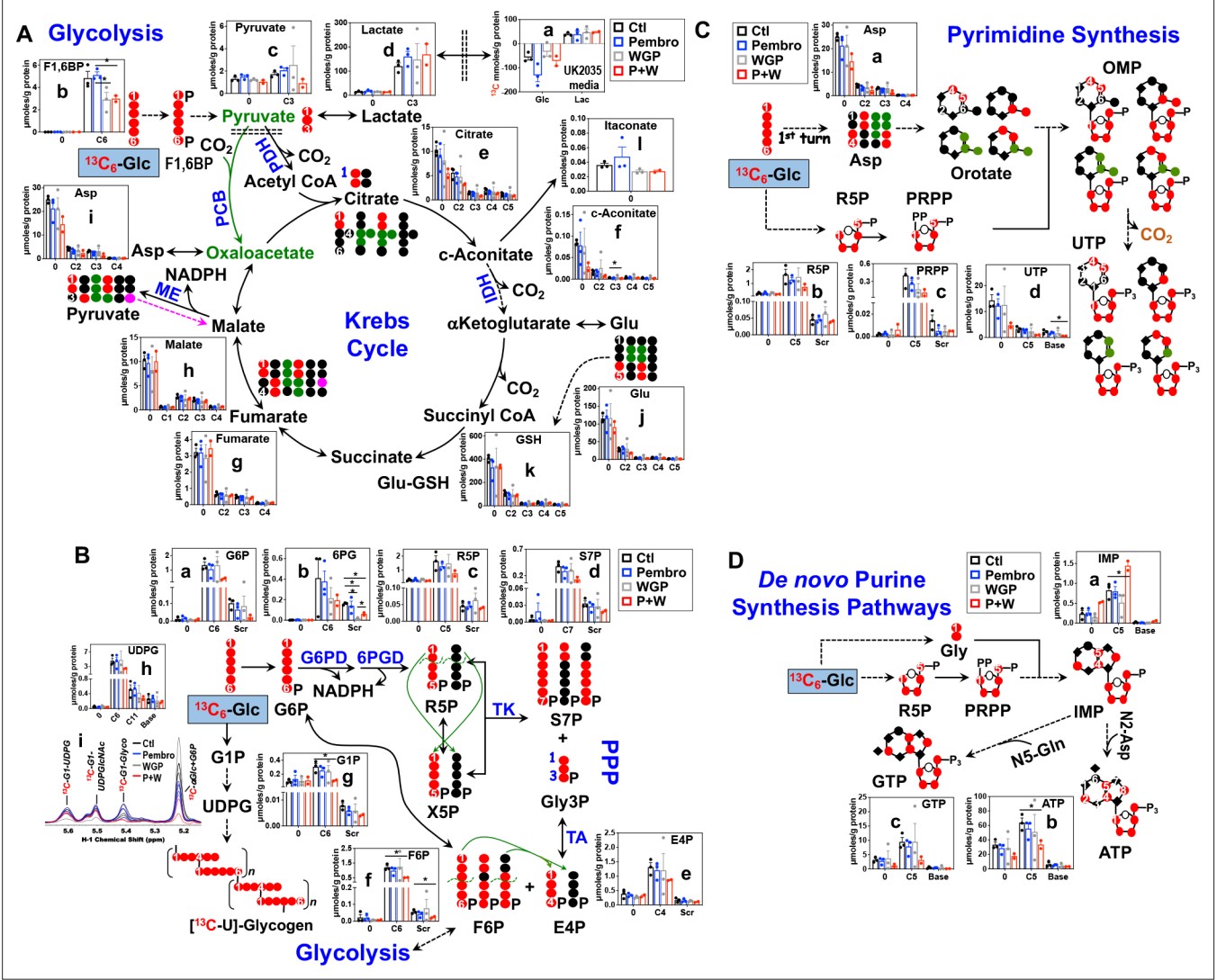

**Figure 5.** Pembro + WGP attenuates central energy and anabolic metabolism in OTCs of brain-metastasized NSCLC tissues from UK2035 patient. CA lung OTCs of UK2035 patient were treated with Ctl (■), 40 µg/mL Pembro (■), 0.1 mg/mL WGP (■) (n = 3), or Pembro + WGP combined (P + W ■) (n = 2) in the presence of $^{13}C_6$-Glc for 24 hr before extraction for polar metabolites and analysis by IC-UHR-FTMS, as described in the Materials and Methods. The diagrams in (**A–D**) depict atom-resolved transformation of $^{13}C_6$-Glc via glycolysis+ the Krebs cycle, the PPP+ glycogen synthesis pathway, and pathways of pyrimidine and purine nucleotide synthesis, respectively. Base in X-axis of (**C**) and (**D**) denotes $^{13}C$-labeled isotopologues of pyrimidine and purine bases. OMP: orotidine 5'-monophosphate; IMP: inosine 5'-monophosphate. All other symbols and abbreviations are as in *Figures 1–2*. Data are displayed as mean ± sem. *p<0.05.

## Pembro+WGP-induced metabolic changes are accompanied by proinflammatory responses and extensive cell death in UK2035 CA lung tissues

The above metabolic disruptions induced by *P + W* treatment were accompanied by increased release of proinflammatory effectors IL-6, INFα, IL-8, and MIP-1α by CA OTC relative to Ctl, Pembro, or WGP treatment (*Figure 6A–D*). Although these changes did not reach statistical significance (p-values ≤ 0.05), the trend points to M1-type polarization elicited by *P + W* treatment.

To relate the metabolic and medium effector changes described above to CA tissue status, we multiplex-stained for PCNA, caspase 3 (CAS), and RIP1 on subsamples of the SIRM tissue OTC. *Figure 6E–H* showed highly heterogenous distribution of PCNA fluorescence (green) for all tissue sections, which presumably reflects the heterogenous distribution of cancer cells. Notably, we observed increased caspase (CAS, orange) and RIP1 (red) fluorescence in *P + W*-treated tissues (**H**) versus Ctl- (**E**), Pembro- (**F**), or WGP (**G**)-treated tissues. Also noted was the higher occurrence of

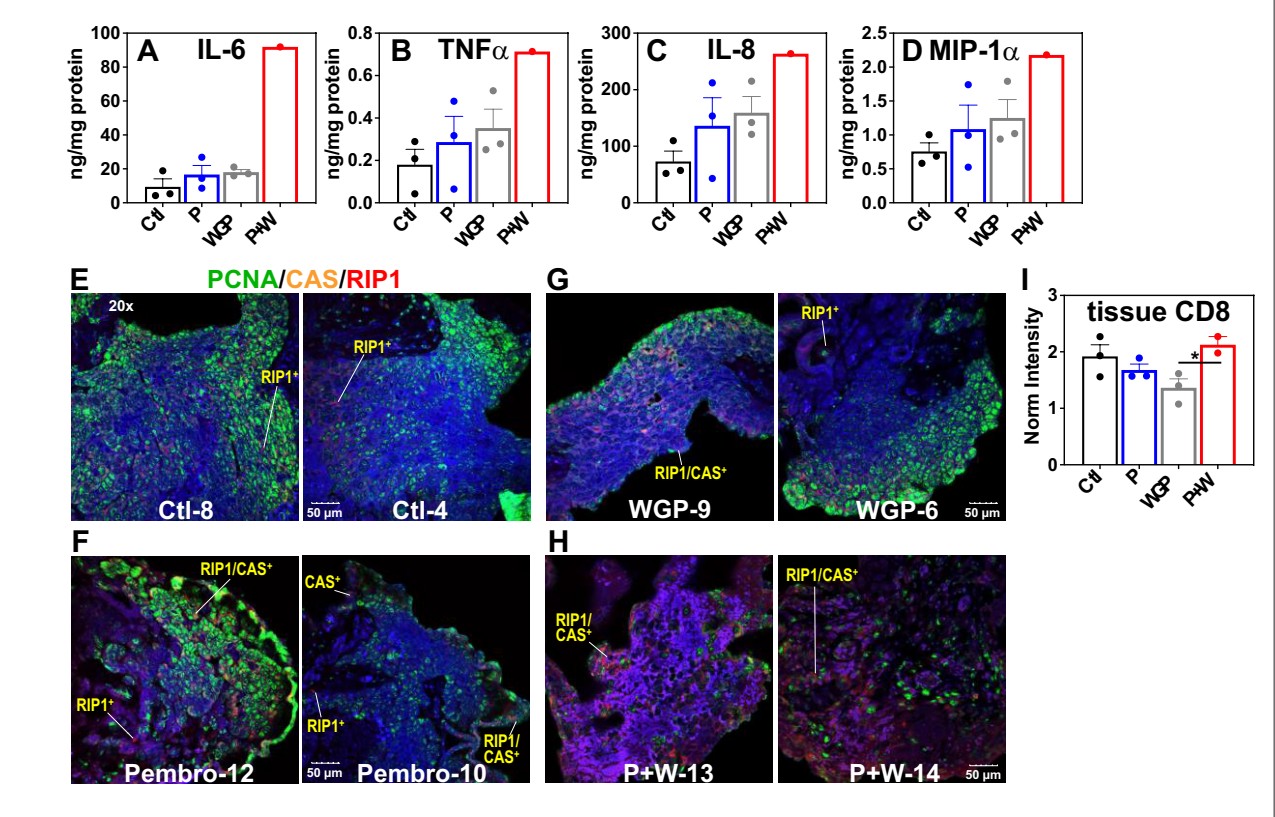

**Figure 6.** *Pembro+ WGP elicits the release of proinflammatory effectors and cell death in UK2035 CA lung tissues.* CA lung OTCs of UK2035 patient were treated with Ctl (■), 40 µg/mL Pembro (■), 0.1 mg/mL WGP (■) (n = 3), or Pembro + WGP combined ($P$ + W ■) (n = 2) in the presence of $^{13}C_6$-Glc for 24 hr. Tissues were subsampled for polar extraction (*Figure 5*) and formaldehyde fixation while media were analyzed for immune effectors (**A–D**). The fixed tissues were FFPE-processed, sectioned as 4 µm slices, stained for PCNA/caspase 3 (CAS)/RIP-1, and analyzed by confocal microscopy (**E–H**), as described in the Experimental. Also shown are the changes in CD8 protein intensity (**I**) in response to treatments, acquired from RPPA as described in the Experimental. Ctl: control; P: Pembrolizumab; WGP: whole glucan particulate; $P$ + W: Pembrolizumab + WG. Panels (**A–D**) display mean ± sem with n = 3. *p<0.05.

CAS⁺/RIP1⁺ cells in $P$ + W-treated tissues than Ctl-, Pembro-, and WGP-treated tissues. These data suggest that $P$ + W-treated OTCs were more apoptotic and necrotic than Ctl-, Pembro-, or WGP-treated OTCs. Such compromised tissue status correlated with the extensive attenuation of central metabolism in $P$ + W-treated OTCs from BM NSCLC tumors (*Figure 5*). This disruption deviated from that seen in Pembro- (*Figure 1*) or WGP-treated (Figures 2 and 4 in *Fan et al., 2016*) primary NSCLC OTCs, except for the block at G6PDH. Moreover, a slightly increased expression of CD8 protein in CA tissues under $P$ + W versus Ctl, Pembro, or WGP treatment was observed (*Figure 6I*), which could reflect T-cell activation in the $P$ + W-treated OTC. Altogether, we surmise that the metabolic attenuation associated with tissue damage overwhelm metabolic stimulation induced by immune activation and that Pembro and WGP synergize in eliciting tissue damage and disrupting tissue metabolism.

## Conclusions

Our studies illustrate the value of patient-derived ex vivo tissue cultures in probing in-depth metabolic and cellular sensitivity to immune modulators in the patient's native TME. We observed metabolic responses to Pembro that are likely to be caused by M1 repolarization of TAM. By coupling with the high-plex DSP analysis, we also saw TMEs likely to be immunoevasive in primary NSCLC tissues of a patient that were overall sensitive to Pembro treatment. We further observed synergism in the tumor-killing action of Pembro and WGP in brain-metastasized NSCLC tissues. Thus, the ex vivo organotypic tissue culture system represents a unique model for interrogating patient's responses to immunotherapeutics while informing potential resistance mechanism in highly heterogeneous TMEs.

# Materials and methods

**Key resources table**

| Reagent type (species) or resource | Designation | Source or reference | Identifiers | Additional information |
|---|---|---|---|---|
| Biological Sample (*Homo sapiens*) | Primary patient NSCLC and matched non cancerous lung tissue | Fresh thin tissue slices from surgical resection | | See Materials and methods section, this study |
| Biological Sample (*Homo sapiens*) | NSCLC metastasis to brain patient | Fresh thin tissue slices from surgical resection | | See Materials and methods section, this study |
| Biological sample (*H. sapiens*) | Primary monocytes | Healthy volunteer | | Freshly isolated from healthy male volunteer |
| Antibody | Pembrolizumab (humanized mouse anti PD-1 monoclonal) | Markey Cancer Center/Merck | | IF(1:1000), WB (1:1000) |
| Antibody | Anti-PanCK (mouse monoclonal) | Invitrogen | *Supplementary file 4* | (1:500) |
| Antibody | Anti-CD8 (mouse monoclonal) | Cell Signaling | *Supplementary file 4* | (1:200) |
| Antibody | Anti-CD68 (mouse monoclonal) | SCBT | *Supplementary file 4* | (1:400) |
| Chemical compound, drug | WGP Beta glucan | InvivoGen | | Whole glucan particles |
| Software, algorithm | MNOVA 14.1 | Mrestlab | | |
| Other | DAPI stain | Invitrogen | D1306 | (1 µg/mL) |
| Other | AlexaFluor Tyramide Superboost kits, | Invitrogen | | See Materials and methods section for detailed use |

## Patient recruitment and tissue procurement/processing

Two surgical patients (UK131 and UK2035) were consented for freshly resected tissue specimens under the approved IRB protocol (14–0288-F6A; 13-LUN-94-MCC) of the University of Kentucky (UKy) (*Supplementary file 1*). Primary cancerous (CA) and surrounding non-cancerous lung tissues of UK131 and brain-metastasized lung cancer tissues of UK2035 were obtained from the surgeons immediately after resection at the operating room. Tissues were transported in DMEM to the laboratory and embedded in 3 % low-melting agarose before sectioning into 750 µm thick slices in PBS under sterile conditions using a Krumdieck tissue slicer (Alabama R&D) according to the vendor's protocol.

## Ex vivo organotypic tissue slice culturing, treatments, and extraction

Tissue slices were cultured for 24 hr in complete DMEM with glucose or Gln replaced by 10 mM $^{13}C_6$-glucose or 2 mM $^{13}C_5,^{15}N_2$-Gln as described previously (*Sellers et al., 2015*). The UK131 tissue slices were co-treated with 40 µg/mL Pembro or vehicle while the UK2035 tissue slices were co-treated with 40 µg/mL Pembro, 0.1 mg/mL β-glucan formulated as whole glucan particulates (WGP soluble, InvivoGen), Pembro and WGP combined, or vehicle. Pembro used was freshly prepared by the UKy Pharmacy, the majority of which was used for NSCLC patient therapy in the clinic. WGP was prepared as described previously (*Fan et al., 2016*). Treatment media were sampled at the start and right before tissue harvest. At harvest, tissues were subsampled for buffered formalin fixing before rapid rinsing three times with excess cold PBS and once with cold nanopore water to remove medium components. Tissues were then flash-frozen in liquid $N_2$ to quench metabolism and stored at −80 °C before further processing. The formalin-fixed tissues were paraffin-embedded (FFPE) and sectioned at 4 µm thickness for H&E and mIF staining.

Frozen tissue slices were pulverized into fine particles using a cryo ball mill (SPEX 6775 Freezer/Mill) before simultaneous extraction for polar and non-polar metabolites using the acetonitrile:H$_2$O:chloroform (2:1.5:1, v/v) solvent partitioning method as described previously (*Fan et al., 2016*). This method also obtained proteins as precipitates, which were solubilized with 62.5 mM Tris + 2 % sodium dodecyl sulfate +1 mM dithiothreitol and denatured at 90 °C for 10 min. Protein concentration in the extracts were determined using the BCA (bicinchoninic acid) method (Pierce Chemicals) to calculate the total

protein content for normalizing the metabolite content. Medium samples were extracted in 80 % cold acetone for polar metabolites as described (*Sellers et al., 2015*).

## Human macrophage culturing, treatment, and extraction

To generate human MΦ, PBMC from a male volunteer age-matched to the cancer patient (65 years old) were isolated using the RosetteSep human monocyte enrichment cocktail kit (StemCell). Isolated monocytes were then differentiated for 5 days in monocytes differentiation medium (MDM) containing DMEM, 10 % FBS, 10 mM glucose, 2 mM Gln, 1 × Anti-anti, and 50 ng/mL CSF-1 at 37 °C/5 % $CO_2$. Seeding density was 12–15 × $10^6$ cells per 10 cm plates. On days 3 of differentiation, media was refreshed with MDM. Naïve MΦ were harvested on day 6 and polarized to either M1 subtype with 100 ng/mL LPS +20 ng/mL IFNγ or to M2 subtype with 20 ng/mL each IL-4+ IL-13 for 2 days. MΦ maintained in MDM served as control (M0). Polarized MΦ were then cultured for 24 hr in MDM medium with glucose or Gln replaced by 10 mM $^{13}C_6$-glucose or 2 mM $^{13}C_5,^{15}N_2$-Gln. Cells were then washed in cold PBS twice and briefly in nanopure water before simultaneous lysing and quenching of metabolism in cold 100 % acetonitrile. Extraction of polar metabolites and proteins in cells and polar metabolites in media were as described above for OTCs.

## Polar metabolite analysis

The polar extracts were split for NMR and/or IC-UHR-FTMS analysis. 1D $^1H$ and $^1H$ (*Albers et al., 2008*) HSQC NMR analyses were performed on a Bruker Avance III 16.45T spectrometer or an Agilent 14.1T DD2 spectrometer outfitted respectively with a 1.7 mm cryoprobe or a 3 mm coldprobe as described previously (*Sellers et al., 2015*). IC-UHR-FTMS analysis was done as described previously (*Fan et al., 2016*) on a Dionex ICS-5000 ion chromatography system coupled to a Thermo Orbitrap Fusion Tribrid Fourier transform mass spectrometer with a mass resolving power of 370,000 at 400 m/z.

## Tissue staining

Four micrometer FFPE sections were stained with hematoxylin and eosin for pathological evaluation. They were also subjected to multiplex immunofluorescence (mIF) staining for protein markers using specific antibodies listed in *Supplementary file 4*,. FFPE sections were deparaffinized, rehydrated, and subjected to heat-induced epitope retrieval by microwaving in Tris–EDTA buffer (10 mM Tris base, 1 mM EDTA solution, 0.05 % Tween 20, pH 9.0) for 10 min at a sub-boiling temperature. They were stained for multiple markers with primary antibodies from the same species using a Tyramide staining method (*Tóth and Mezey, 2007*) according to vendor's protocol (AlexaFluor Tyramide Superboost kits, Invitrogen). Briefly, tissue sections were blocked in 3 % hydrogen peroxide for 10 min at room temperature, washed in PBS, blocked in 10 % goat serum for 1 hr at room temperature, incubated with appropriately diluted primary antibody overnight at 4 °C, washed in PBS, incubated in poly-HRP-conjugated secondary antibody, and washed again in PBS before incubation in a tyramide working solution (e.g. AlexaFluor 647 tyramide) for 10 min followed by immediate application of reaction stop reagent working solution. Tissue sections were then rinsed three times in PBS before microwaving in citrate buffer (10 mM sodium citrate, 0.05 % Tween 20, pH 6.0) for 10 min at a sub-boiling tempera-ture and allowed to cool to room temperature while being kept in the citrate buffer. The samples were then washed twice in PBS, blocked in 3 % hydrogen peroxide and 10 % goat serum again before the application of a second round of primary and secondary antibody, followed by treatment with a second tyramide working solution (e.g. AlexaFluor 594 tyramide). For triple protein marker staining, the process was repeated once more with a third tyramide working solution (e.g. AlexaFluor 488 tyramide). Slides were mounted in Prolong Gold Antifade Mountant with DAPI and dried overnight at 4 °C (Thermo Fisher Scientific, P36935) before confocal microscopic imaging.

## Digital spatial profiling

DSP was performed on one slide each of the FFPE sections for control and Pembro-treated UK131 tissue slices using the GeoMx service at Nanostring. Briefly, tissue slides were first stained for cell lineage markers panCK (cancer cells), CD8 (cytotoxic T cells), and CD68 (MΦ) using fluorescent antibodies (*Supplementary file 4*) and for nuclei using DAPI. Twelve region of interests (ROIs) were then selected for each slide based on the enrichment of panCK, CD8, CD68 and mixed fluores-cence. The slides were incubated with antibodies conjugated with photocleavable oligonucleotide

barcodes (Oligos) to probe a panel of 58 markers for immune functions and cancer cell type/status. The Oligo barcodes from the chosen ROIs were liberated by laser irradiation and quantified by nCounter (Nanostring). The Oligo count for each marker was normalized by the geometric mean of housekeeping genes (GAPDH, S6, and Histone H3). Differential abundance analysis between Ctl and Pembro treatments was performed by limma (*Smyth, 2004*) on $\log_2$-transformed data. Significantly differentially abundant markers were identified at false discovery rate < 0.05. Additional ratioing of the Oligo counts for different markers was performed to reveal activation status of immune cells.

### Statistical analyses

The unpaired two-tailed t-test was used for comparing means, and the raw p-value was corrected for multiple comparisons using the Benjamini–Hochberg procedure (*Benjamini and Hochberg, 1995*) when relevant, accepting q < 0.05 as statistically significant.

## Acknowledgements

We thank Drs. Jessica Macedo and Qiushi Sun for assistance in the MS analysis, Dr. Salim S El-Amouri for assistance in the human macrophage experiment, Dr. Timothy L Scott for tissue slicing, and Ms. Yan Zhang, Teresa Cassel, Hui Lu for assistance in sample processing/extraction and Dr. Penghui Lin for NMR data collection. This work was supported by grants 1P01CA163223-01A1, 1U24DK097215-01A1, 5P20GM121327 and Shared Resource(s) of the University of Kentucky Markey Cancer Center P30CA177558.

## Additional information

### Funding

| Funder | Grant reference number | Author |
| --- | --- | --- |
| University of Louisville | 1P01CA163223-01A1 | Andrew N Lane<br>Teresa WM Fan |
| University of Louisville | 1U24DK097215-01A1 | Richard M Higashi<br>Teresa WM Fan<br>Andrew N Lane |
| University of Louisville | 5P20GM121327 | Andrew N Lane |

The funders had no role in study design, data collection and interpretation, or the decision to submit the work for publication.

### Author contributions

Teresa WM Fan, Conceptualization, Data curation, Formal analysis, Funding acquisition, Investigation, Project administration, Supervision, Validation, Writing – original draft, Writing – review and editing; Richard M Higashi, Data curation, Formal analysis, Funding acquisition, Methodology, Supervision, Writing – review and editing; Huan Song, Formal analysis, Writing – review and editing; Saeed Daneshmandi, Angela L Mahan, Matthew S Purdom, Therese J Bocklage, Thomas A Pittman, Daheng He, Formal analysis, Investigation, Writing – review and editing; Chi Wang, Data curation, Formal analysis, Writing – review and editing; Andrew N Lane, Conceptualization, Data curation, Formal analysis, Funding acquisition, Investigation, Methodology, Project administration, Supervision, Validation, Writing – original draft, Writing – review and editing

### Author ORCIDs

Teresa WM Fan (ID) http://orcid.org/0000-0002-7292-8938
Andrew N Lane (ID) http://orcid.org/0000-0003-1121-5106

### Ethics

Surgical patients were consented for freshly resected tissue specimens under the approved IRB protocol (14-0288-F6A; 13-LUN-94-MCC) of the University of Kentucky (UKy).

**Decision letter and Author response**
Decision letter https://doi.org/10.7554/eLife.69578.sa1
Author response https://doi.org/10.7554/eLife.69578.sa2

## Additional files

### Supplementary files

• Transparent reporting form

• Supplementary file 1. Table S1. Patient demographics and clinical characteristics.

• Supplementary file 2. Primary NSCLC tissue of UK131 patient stains positive forsquamous cell carcinoma marker, PD-1 and PD-L1. Freshly resected CA lung tissue ofUK131 was FFPE-processed, sectioned as 4 µm slices, stained for KRT5 (squamous cellcarcinoma marker)/HIF1α in A, PD-1/CD8 in B, and PD-L1/CD206 in C, and analyzed byconfocal microscopy as described in Materials and methods.

• Supplementary file 3. Brain-metastasized NSCLC tissue of UK2035 patient stains positivefor neuroendocrine markers, PD-1 and PD-L1. Freshly resected CA lung tissue of UK2035was FFPE-processed, sectioned as 4 µm slices, stained for CgA/NCAM1 (neuroendocrinetumor markers) in A and PD-1/PD-L1/CD8 in B, and analyzed by confocal microscopy, asdescribed in the Materials and Methods.

• Supplementary file 4. Table S2. Information on primary antibodies used forimmunofluorescent staining.

### Data availability

All data generated or analysed during this study are included in the manuscript and supporting files. Excel spreadsheets of data used for tables and figures have been deposited at Dryad.

The following dataset was generated:

| Author(s) | Year | Dataset title | Dataset URL | Database and Identifier |
|---|---|---|---|---|
| Lane AN, Fan TWM, Higashi RM, Song H, Daneshmandi S, Mahan AL, Purdom MS, Pittman TA, He D, Wang C | 2021 | Innate immune activation by checkpoint inhibition in patient-derived lung cancer tissues | http://dx.doi.org/10.5061/dryad.n5tb2rbwc | Dryad Digital Repository, 10.5061/dryad.n5tb2rbwc |

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
