## [Decision Letter]

Congratulations, we are pleased to inform you that your article, "Innate immune activation by checkpoint inhibition in patient-derived lung cancer tissues", has been accepted for publication in eLife. Your article has been reviewed by three peer reviewers, one of whom is a member of our Board of Reviewing Editors, and the evaluation has been overseen by a Senior Editor.

*Reviewer #1 :*

Anti-PD-1 monoclonal antibody therapy can influence the survival of lung cancer patients, but the results are variable and not always satisfactory.

The approach is limited by a general lack of information about the activities induced or intercepted by the antibody on the target tumor and - importantly - within the surrounding environment.

The strength of the paper by Fan and Lane is the adoption of ex-vivo tissue slices from primary and metastatic lung tumors. This makes it possible to track the metabolic reprogramming observed in vivo by analyzing metabolic and morphologic studies. Pembrolizumab was used in combination with the natural immune activator β-glycan. The conclusion of the authors may be justified by the reading of their data, at least from the two patients analyzed.

In my view, the most interesting finding was the metabolic shift induced by Pembro + β-glycan, while the analysis of the effects induced on different cell populations is less strong .

I believe that the strength of the paper could be improved by including or discussing the interplay between PD-1 and CD38 (Chen L et al., Cancer Discovery, 2018).

Comparative analysis could be flanked by additional characterization of the populations observed in the primary and secondary tissue sections (in my view, it is difficult to say that CD25 is a marker of Treg).

Another simple adjunct would be the inclusion of the analysis of the expression of adenosine receptors (Koness J. M. et al, Cells, 2019).

In conclusion, the authors should be commended for their brilliant paper and for their choice of analytical technologies.

*Reviewer #2:*

The role of immunotherapy rapidly gained a transformative impact in cancer therapy. Nevertheless, modeling the alteration induced by checkpoint inhibitors (e.g. pembrolizumab) is essential to stratify the various degree of response and predict the onset of resistance. To this aim, it is important to develop novel models to integrate existing ones. In this work, Fan et al characterized and refined the tissue slides (originally used by Warburg for his seminal observation in cancer metabolism) by culturing primary tumor sections embedded in agarose, as a tool for modeling the response to checkpoint inhibition while preserving tissue architecture and heterogeneity.

Strength

The model is original and took advantage of a system which is certainly under investigated. Moreover, the techniques used to study heterogeneity as well the metabolomic approach are certainly compelling and provide an accurate indication of the complexity in a tumor model.

Fig4 is very informative providing some insight of the different immune landscape alteration on a slide

Weakness

The potential of this approach is certainly elevated. Nevertheless, the work fails to present clear differences between treatments, for instance:

Fig1 I am not able to see an activation of PPP, besides the increase in S7P, no significant regulation following Pembro treatment were evident.

Fig2 likewise, although the tracing experiment is very elegant, very limited alterations is present besides tumor/non tumor tissue.

Fig3 No biologically relevant alteration is present, plus it is not clear to understand what are the legends presented (C-Glc P-Gln...)?

The rationale behind the use of WGP in combo should be introduced more explicitly, with reference for the relevance in therapy. More data on the response in patients (if not available from the same patient also in similar treatment)

Overall, it is difficult correlate the metabolic rewiring of the macrophages, as reported in Fig S3 (consistent with data already published), with the effects of Pembro-treatment on whole tumor slide.

This work presents various critical points which undermine its significance and makes many of the conclusion non convincing:

The impact of this present work is limited by the use of only two tumor samples.

Only 2 different tissues are not informative: at least these results should have been compared with primary tumor analysis and/or with murine transposition of the same approach and/or with more clinical data from patients.

Moreover, since the authors state in the conclusions paragraph "a patient that was overall sensitive to Pembro treatment", more detailed clinical data should be added to understand if the primary OTCs are really predictive models. Overall, a discussion section is necessary to improve the quality of the manuscript.

Fig5 it is probably a mistake of figure reporting, but the only treatment which appears with great interest (Pembro + WGP) consists of a single data point.

Overall, it is difficult correlate the metabolic rewiring of the macrophages, as reported in Fig S3 (consistent with data already published), with the effects of Pembro-treatment on whole tumor slide. The metabolic profile of the tumor slides cannot represent the metabolic changes of macrophages (5% of density).

In order to improve the quality of the paper and further support their statement the authors should perform an ex vivo treatment of fresh tumour tissues [10.1038/s41598-017-12222-9] and follow with FACS sorting [10.7554/eLife.61980] coupled with metabolomic analysis.

Moreover, a NanoString approach such as nCounter Metabolic Pathways Panel coupled with GeoMx Digital Spatial Profiler could strengthen the metabolomic analysis and be useful to build a metabolic signature predictive for ICIs response.

Tissue slices should be cultured in a more physiological medium (i.e. Plasmax) instead of DMEM.

*Reviewer #3:*

In the present paper the Authors aimed at investigating the impact of immune checkpoint inhibition in the modulation of cancer cell metabolism and in the impairment of the overall immune tumor microenvironment.

Major strengths of the paper are the innovative model and the study design, aimed at the localization and quantification "in situ" of the set of molecules under investigation.

Major weaknesses are the limited statistical significance of their data and the heterogeneity of the model (poorly characterized tumor typing of both primary tumor and brain metastasis and different treatment approaches in the two models).

Based on strengths and weaknesses above, the Authors partially achieved their objectives, and the conclusions have to be supported in a validation set of experiments. This is of paramount importance for the translational impact of their research. In fact, their model has a clear benefit in helping to understand the impact of immune checkpoint molecules (i.e. pembrolizumab) in the modulation of the immune-to-tumor response, but their data are largely sample-specific and do not take into account the extreme heterogeneity among different cancers.

– The histology of the two tumor samples is not well characterized, despite it is well known that immune-to-tumor response mechanisms are largely heterogeneous among different non-small cell lung cancer histotypes. CK5 expression is not enough to state squamous cell carcinoma (better markers might be p40/p63) and the terminology of the metastatic tumor is not clear if refers to "carcinoid" or "large cell neuroendocrine carcinoma" (much more likely in the presence of brain metastasis). The Authors may come back to the original diagnoses of their samples, with special reference to the metastatic one.

– Statistical significance is reached for a minority of experiments, only, whereas the majority of data are described as significant but with no significant p values. As an example, in figure 1 S7P, only, reached statistical significance whereas in Figure 2 I did not find any statistically significant result. Moreover, as a comment to Figure 3 the Authors state that the majority of investigated molecules did not reach statistical significance, but I could not find the minority with a significance.

– The data reported in Figure 4A and 4B are descriptive, but it is not clear if they refer to the same tumor areas in serial sections (as they seem to be different in the HandE panels). The same applies to Figure 5 E to H. Moreover, a quantitative or semiquantitative estimation of the markers should be performed to sustain significance of variations reported. Data illustrated in Figure 4C are also biased by the different areas evaluated. Although the Authors selected fields with enrichment of tumor cells they loose direct comparison of same tumor areas.

– The Authors might comment of the increased MO content in CA-Pem (figure 4A) and the reduced CD68 expression illustrated in figure 4D.

– It is not clear what the reference in lines 141-142 is related to.

---

## [Author Response]

Reviewer #1:Anti-PD-1 monoclonal antibody therapy can influence the survival of lung cancer patients, but the results are variable and not always satisfactory.The approach is limited by a general lack of information about the activities induced or intercepted by the antibody on the target tumor and - importantly - within the surrounding environment.The strength of the paper by Fan and Lane is the adoption of ex-vivo tissue slices from primary and metastatic lung tumors. This makes it possible to track the metabolic reprogramming observed in vivo by analyzing metabolic and morphologic studies. Pembrolizumab was used in combination with the natural immune activator β-glycan. The conclusion of the authors may be justified by the reading of their data, at least from the two patients analyzed.In my view, the most interesting finding was the metabolic shift induced by Pembro + β-glycan, while the analysis of the effects induced on different cell populations is less strong .

The patient-derived tissue slices maintained the original tumor architecture, which varied throughout the tumor bulk. This heterogeneity is what we sought to demonstrate in this paper. We agree that the approach taken did not directly provide changes of metabolic activity in individual cell types in the tumor microenvironment (TME). However, by relating metabolic/immune functional changes of macrophages in vitro to those observed ex vivo, we can tease apart the overall changes in tumor associated macrophages (TAM) or cancer cells in the native TME in response to immune modulators with the in-depth analysis we opted for in this article.

I believe that the strength of the paper could be improved by including or discussing the interplay between PD-1 and CD38 (Chen L et al., Cancer Discovery, 2018).

This interplay could occur in our case. However, we did not analyze for CD38 in our samples and it would be too speculative to discuss this interplay. However, we observed T cell activation (as evidenced by relatively high granzyme B expression and iCOS/CD4 ratio) in one resistant TME region (ROI 8), so it is unlikely that the resistance was mediated by such interplay in this region.

Comparative analysis could be flanked by additional characterization of the populations observed in the primary and secondary tissue sections (in my view, it is difficult to say that CD25 is a marker of Treg).

It would be ideal to get primary and secondary tumors from the same patient, but to date this has not been possible. We agree that there are multiple markers to define human Treg (CD3^+^, CD4^+^, CD25^+^, CD127^+^, FoxP3^+^). Further T cell marker analysis is warranted to substantiate this point but is beyond the scope of this paper.

Another simple adjunct would be the inclusion of the analysis of the expression of adenosine receptors (Koness J. M. et al, Cells, 2019).

We thank the reviewer for this suggestion, and we will include this in our follow-up study.

In conclusion, the authors should be commended for their brilliant paper and for their choice of analytical technologies.

The reviewer’s comment is much appreciated.

Reviewer #2:The role of immunotherapy rapidly gained a transformative impact in cancer therapy. Nevertheless, modeling the alteration induced by checkpoint inhibitors (e.g. pembrolizumab) is essential to stratify the various degree of response and predict the onset of resistance. To this aim, it is important to develop novel models to integrate existing ones. In this work, Fan et al characterized and refined the tissue slides (originally used by Warburg for his seminal observation in cancer metabolism) by culturing primary tumor sections embedded in agarose, as a tool for modeling the response to checkpoint inhibition while preserving tissue architecture and heterogeneity.

We thank the reviewer for the positive comment. The tissues were sliced in agarose but were incubated as free cultures for metabolic studies.

StrengthThe model is original and took advantage of a system which is certainly under investigated. Moreover, the techniques used to study heterogeneity as well the metabolomic approach are certainly compelling and provide an accurate indication of the complexity in a tumor model.Fig4 is very informative providing some insight of the different immune landscape alteration on a slide

We thank the reviewer for these positive comments.

WeaknessThe potential of this approach is certainly elevated. Nevertheless, the work fails to present clear differences between treatments, for instance:Fig1 I am not able to see an activation of PPP, besides the increase in S7P, no significant regulation following Pembro treatment were evident

As we pointed out in the paper, pembrolizumab-induced changes of individual metabolites in a given pathway such as PPP in Fig. 1 did not reach statistical significance, at least in part due to intrinsic tissue heterogeneity. However, there were consistent changes in the majority of metabolites in the pathways depicted in the tumor (but not in the control) tissues, which points to altered pathway activity.

Fig2 likewise, although the tracing experiment is very elegant, very limited alterations is present besides tumor/non tumor tissue.

Please see response to the comment above. In particular, the tumor OTCs but not the non-cancerous counterparts showed consistent changes in pyruvate and all Krebs cycle metabolites, which points to altered Krebs cycle activity.

Fig3 No biologically relevant alteration is present, plus it is not clear to understand what are the legends presented (C-Glc P-Gln...)?

The data shown are changes in cytokines released into the media, which reflects changes in immune functions and crucial to assessing Pembro’s effect. The abbreviated texts were defined in the Fig. 3’s legend as C-Glc or C-Gln: Ctl + ^13^C_6_-Glc or ^13^C_5,_^15^N_2_-Gln; P-Glc or P-Gln: 40 µg/mL Pembro + ^13^C_6_-Glc or ^13^C_5,_^15^N_2_-Gln.

The rationale behind the use of WGP in combo should be introduced more explicitly, with reference for the relevance in therapy. More data on the response in patients (if not available from the same patient also in similar treatment)

WGP and Pembrolizumab are known to activate different compartments of the immune system, i.e. macrophages and T cells, for which we have provided relevant references in the original text. This was the basis to combine the two treatments to see if they act synergistically in immune activation and tumor killing.

Overall, it is difficult correlate the metabolic rewiring of the macrophages, as reported in Fig S3 (consistent with data already published), with the effects of Pembro-treatment on whole tumor slide.

Figure S3 (now Figure 1- figure supplemental 1) depicted the metabolic reprogramming of isolated human macrophages in response to pro-inflammatory stimuli, which was comparable to that of ex vivo patient-derived OTCs in response to Pembro treatment. This comparison enabled us to discern overall TAM response to Pembro in whole tumor slices. In situ single-cell based SIRM analysis will provide the ultimate proof for this but is beyond current technological limits.

This work presents various critical points which undermine its significance and makes many of the conclusion non convincing:The impact of this present work is limited by the use of only two tumor samples.Only 2 different tissues are not informative: at least these results should have been compared with primary tumor analysis and/or with murine transposition of the same approach and/or with more clinical data from patients.Moreover, since the authors state in the conclusions paragraph "a patient that was overall sensitive to Pembro treatment", more detailed clinical data should be added to understand if the primary OTCs are really predictive models. Overall, a discussion section is necessary to improve the quality of the manuscript.

We agree that more examples will provide additional information on how variably patient tissues can respond to immune modulations, which is part of our ongoing studies. However, each patient’s TME is unique in its own right, which provides insights into individual patient’s drug response without the need for additional comparison. In addition, the native TME and tumor architecture will be lost in the murine explant (PDX) model, which deviates from the purpose of the study. The paragraph cited in the Conclusions section was meant for overall sensitivity of the tumor OTCs, not of the patient. Since Pembro had not been approved for treating primary respectable NSCLC, we do not have the correlation of OTCs response to patient response. This is certainly of interest to pursue in future studies.

Fig5 it is probably a mistake of figure reporting, but the only treatment which appears with great interest (Pembro + WGP) consists of a single data point.

It is not a mistake. We have only one medium sample from the Pembro + WGP treatment available for the cytokine analysis.

Overall, it is difficult correlate the metabolic rewiring of the macrophages, as reported in Fig S3 (consistent with data already published), with the effects of Pembro-treatment on whole tumor slide. The metabolic profile of the tumor slides cannot represent the metabolic changes of macrophages (5% of density).

Please see response to comment 7 on Fig. S3 above. Although we did not quantify the % of macrophages in tumor tissue slices, it is higher than 5% of tumor cells based on the CD68 and panCK staining patterns in Fig. 4C. We would also like to point out that macrophages are much bigger cells than lymphocytes and isolated macrophages show very large metabolic responses to proinflammatory (M1) activation (see Figs. S3 and S4; now Fig.1 – figure supplement 1 and Fig.2 – figure supplement 1). These responses are expected to be “diluted” in patient OTCs by the presence of other cell types, foremost cancer cells. This is consistent with the much less prominent M1-like responses in cancer OTCs.

In order to improve the quality of the paper and further support their statement the authors should perform an ex vivo treatment of fresh tumour tissues [10.1038/s41598-017-12222-9] and follow with FACS sorting [10.7554/eLife.61980] coupled with metabolomic analysis.

We thank the reviewer for the suggestion. However, tissue dissociation followed by FACS sorting is known to alter cell metabolite patterns. We would expect that the tracer enrichment patterns will be even more significantly altered or even lost during the prolonged harvest procedure.

Moreover, a NanoString approach such as nCounter Metabolic Pathways Panel coupled with GeoMx Digital Spatial Profiler could strengthen the metabolomic analysis and be useful to build a metabolic signature predictive for ICIs response.

We agreed that coupling RNA expression analysis with DSP and SIRM analysis would further strengthen the paper. However, we did not have additional OTCs to perform parallel RNA expression analysis, as the fresh patient tissues that we could procure were only sufficient for SIRM study.

Tissue slices should be cultured in a more physiological medium (i.e. Plasmax) instead of DMEM.

We agree that standard media such as DMEM do not represent typical physiological conditions and new medium formulations such as Plasmax more closely mimic metabolite composition of human serum. We will take this into consideration in future studies. In fact, we are exploring different culturing methods to prolong the viability of OTCs such that chronic (e.g. month-long) drug response study can be performed. We would like to point out that the metabolite composition of serum does vary from patient to patient, which makes it difficult to match with a common medium. We also need to supplement certain growth factors to maintain cancer stem cells in OTCs, which are crucial to drug resistance development. Moreover, unlike 2D cell cultures, we expect OTCs to develop localized gradient of nutrients, pH, and waste products such as lactate, which makes complete matching of the metabolite composition of the TME more complex. Suffice to say, we observed metabolic/immune functional responses of OTCs to immune modulators that were consistent with macrophage activation, regardless the use of unphysiological DMEM.

Reviewer #3:In the present paper the Authors aimed at investigating the impact of immune checkpoint inhibition in the modulation of cancer cell metabolism and in the impairment of the overall immune tumor microenvironment.

We thank the reviewer for the positive comments.

Major strengths of the paper are the innovative model and the study design, aimed at the localization and quantification "in situ" of the set of molecules under investigation.Major weaknesses are the limited statistical significance of their data and the heterogeneity of the model (poorly characterized tumor typing of both primary tumor and brain metastasis and different treatment approaches in the two models).Based on strengths and weaknesses above, the Authors partially achieved their objectives, and the conclusions have to be supported in a validation set of experiments. This is of paramount importance for the translational impact of their research. In fact, their model has a clear benefit in helping to understand the impact of immune checkpoint molecules (i.e. pembrolizumab) in the modulation of the immune-to-tumor response, but their data are largely sample-specific and do not take into account the extreme heterogeneity among different cancers.

Please also see response to Reviewer 2’s comment 3. We agree that individual metabolite responses did not reach statistical significance, presumably due to the influence of intrinsic heterogeneity in OTCs. However, metabolites from the same pathway responded in the same direction, which points to altered pathway activity. Patient tissue-specific response to drugs under the influence of tissue heterogeneity is what we sought to investigate. We illustrated this with two patient examples and did not intend to draw conclusion on how NSCLC tissues respond to Pembro or WGP in general. We are continuing this type of study on other patients’ OTCs and will follow reporting many more patient examples to further address the tumor heterogeneity issue.

– The histology of the two tumor samples is not well characterized, despite it is well known that immune-to-tumor response mechanisms are largely heterogeneous among different non-small cell lung cancer histotypes. CK5 expression is not enough to state squamous cell carcinoma (better markers might be p40/p63) and the terminology of the metastatic tumor is not clear if refers to "carcinoid" or "large cell neuroendocrine carcinoma" (much more likely in the presence of brain metastasis). The Authors may come back to the original diagnoses of their samples, with special reference to the metastatic one.

The tumor classification was based on certified clinical diagnosis and histological examination by co-authors Drs. Bocklage and Purdom who are professional pathologists. The histochemical marker selection was according to the pathologist’s suggestion to corroborate with the clinical diagnosis.

– Statistical significance is reached for a minority of experiments, only, whereas the majority of data are described as significant but with no significant p values. As an example, in figure 1 S7P, only, reached statistical significance whereas in Figure 2 I did not find any statistically significant result. Moreover, as a comment to Figure 3 the Authors state that the majority of investigated molecules did not reach statistical significance, but I could not find the minority with a significance.

Please see also response to Comment 2. We did not provide the p values for the majority of metabolites in Figs, 1, 2, and S6 (now Fig. 5) as they were greater than 0.05. We apologize for the oversight of stating “the majority of Pembro effects on CA OTC did not reach statistical significance” for Fig. 3. It should be “none of Pembro effects on CA OTC reach statistical significance”. We have modified the text accordingly.

– The data reported in Figure 4A and 4B are descriptive, but it is not clear if they refer to the same tumor areas in serial sections (as they seem to be different in the HandE panels). The same applies to Figure 5 E to H. Moreover, a quantitative or semiquantitative estimation of the markers should be performed to sustain significance of variations reported. Data illustrated in Figure 4C are also biased by the different areas evaluated. Although the Authors selected fields with enrichment of tumor cells they lose direct comparison of same tumor areas.

The slices in Figs. 4 and 5 (now Fig. 6) were different but prepared from the same block of patient tissue. The distributions of cells varied from slice to slice, reflecting bulk tissue heterogeneity and giving rise to somewhat different staining patterns. The quantitative image analysis for Fig. 4B was unreliable due to large and variable autofluorescence background in different regions of the stained slides. We did obtain quantitative marker data with DSP analysis of panel 4C, which were shown in panels D and E. DSP analysis was not affected by autofluorescence. Choosing different ROIs is how DSP is done such that different ROIs represent different TME. As stated on p. 8 of the revised text, our ROI selections were meant to compare different TMEs, which was enriched in cancer cells, macrophages, lymphocytes or mixed cell types. This is not a bias, rather the selections enabled comparison of different TMEs’ response to Pembro.

– The Authors might comment of the increased MO content in CA-Pem (figure 4A) and the reduced CD68 expression illustrated in figure 4D.

We did not quantify the macrophage content in Fig. 4A, so we are not sure how the reviewer came up with the “increased MO content in CA-Pem”. In fact, by visual examination, the macrophage content may have decreased in CA-Pem

– It is not clear what the reference in lines 141-142 is related to.

The Kashiwagi reference refers to the molecular markers for diagnosing lung neuroendocrine tumors (LNETs).